# Mapping the HPV Landscape in South African Women: A Systematic Review and Meta-Analysis of Viral Genotypes, Microbiota, and Immune Signals

**DOI:** 10.3390/v16121893

**Published:** 2024-12-08

**Authors:** Carol K. Maswanganye, Pamela P. Mkhize, Nontokozo D. Matume

**Affiliations:** 1Discipline of Genetics, School of Life Sciences, University of KwaZulu-Natal, Pietermaritzburg 3209, South Africa; 223020231@stu.ukzn.ac.za; 2Discipline of Biochemistry, School of Life Sciences, University of KwaZulu-Natal, Pietermaritzburg 3209, South Africa; mkhizep6@ukzn.ac.za; 3Centre for the AIDS Programme of Research in South Africa (CAPRISA), University of KwaZulu-Natal Medical School, Durban 4013, South Africa; 4SAMRC-UNIVEN Antimicrobial Resistance and Global Health Research Unit, HIV/AIDS & Global Health Research Programme, University of Venda, Thohoyandou 0950, South Africa

**Keywords:** human papillomavirus (HPV), prevalence, genotype, cytokines, cervicovaginal microbiota, cervical cancer

## Abstract

This systematic review and meta-analysis evaluate human papillomavirus (HPV) prevalence, genotype distribution, and associations with cervicovaginal microbiota and cytokine profiles among South African women, where cervical cancer ranks as the second most common cancer. PubMed, SCOPUS, and Web of Science were searched for studies on HPV infection up to 21 September 2024. The pooled prevalence was estimated using a random-effects model, with subgroup analyses by province, sample type, and HIV status. Publication bias was evaluated using funnel plots and Egger’s test. Of the 19,765 studies screened, 120 met the inclusion criteria, comprising 83,266 participants. Results indicate a high HPV burden, with a pooled prevalence of 58% (95% CI: 52–64%), varying regionally from 53% (95% CI: 41–65%) to 64% (95% CI: 55–73%), with some regions under-researched. Cervical samples had the highest HPV prevalence (60% (95% CI: 54–66%)), while non-genital samples were less studied. High-risk (HR) HPV types, notably HPV 16 (7.5%), HPV 35 (4.1%), and HPV 18 (3.9%), were prominent, with HPV 35 emphasizing the need for expanded vaccine coverage. HIV-positive women had a higher pooled HPV prevalence (63% (95% CI: 55–71%)). Funnel plot analysis and Egger’s test suggested a potential publication bias (*p* = 0.047). HPV-positive women exhibited lower Lactobacillus levels and an increase in *Bacterial Vaginosis* (BV)-associated species like *Gardnerella*, potentially supporting HPV persistence. Cytokine analysis showed elevated MIP-1α and MIP-1β in HPV infections, though cytokine profiles may depend on HPV genotypes. These findings underscore the need for research on HPV–microbiome-immune interactions and call for comprehensive HPV-prevention strategies, including vaccines targeting regional HPV types and tailored interventions for HIV-positive populations.

## 1. Introduction

Human papillomavirus (HPV) is predominantly transmitted through sexual contact and targets basal keratinocytes in genital mucosa, oral mucosa, and skin. Over 200 distinct HPV genotypes exist, categorised based on their carcinogenic potential as low risk (LR) and high risk (HR) [1]. Genital warts are caused by LR-HPV types, while persistent HR-HPV types are associated with precancerous lesions and found in 99.7% of cervical malignancies globally [2]. Invasive cervical cancer (ICC) represents a significant global health burden, with an age-standardised incidence rate of 13.3 per 100,000 in 2020 [3]. This cancer disproportionately affects women in developing countries, with evidence suggesting a rising incidence trend in sub-Saharan Africa [4,5]. South Africa is one of the countries in sub-Saharan Africa with the highest rates of cervical cancer incidence, making ICC a major leading cause of cancer-related deaths among women [6].

HPV is not only responsible for cervical cancer but also implicated in other anatomical areas, including the anus and oral cavity, where it has been identified as a significant etiological factor in anal and oropharyngeal cancers [7,8,9,10]. Oncogenic HPV types are estimated to cause 90% of anal squamous cancers [11]. Approximately 30% of oral cancer cases are linked to HPV, with most other cases associated with smoking and alcohol use [12]. While HPV infections in the oral and anal regions occur less frequently than cervical infections, they are more common among individuals with compromised immune systems, including those living with human immunodeficiency virus (HIV) [13]. Previous studies have linked these cancers to HR-HPV-16 [11,14]. However, current epidemiological and molecular evidence supporting a causal relationship remains limited, as it primarily revolves around detecting HPV genomes in tumour cells without more definitive conclusions. Data on oral and anal HPV infections in African settings are scarce, highlighting a significant gap in understanding the role of HPV in these cancers within this population.

In South African women, HR-HPV genotypes, notably HPV-16 and HPV-18, along with LR-HPV types such as HPV-6 and HPV-11, are predominant, reflecting global patterns [15]. Other significant HR-HPV types including HPV-31, HPV-33, HPV-35, HPV-39, HPV-45, HPV-51, HPV-52, HPV-56, HPV-58, HPV-66, and HPV-68 [16,17,18] are at an increasing rate globally. HPV-35, HPV-39, HPV-51, HPV-56, HPV-59, HPV-66, and HPV-68 are more prevalent in Africa than elsewhere and are not covered by existing vaccines. HPV-35 is detected in around 10% of cervical cancer cases in Africa, compared to its detection in about 2% globally [19]. Currently, three HPV vaccines are commercially available: Cervarix^®^, Gardasil^®^, and Gardasil^®^9 [20]. The bivalent vaccine Cervarix^®^ targets the two most prevalent HR-HPV types, HPV-16 and HPV-18, while Gardasil^®^, a quadrivalent vaccine protects against both HR types (HPV-16 and 18) as well as two LR types (HPV-6 and 11) that cause genital warts [21]. The nonavalent vaccine Gardasil^®^9 offers broader protection by targeting nine HPV genotypes, covering four types from the bivalent and quadrivalent vaccines (HPV-16, 18, 6, and 11) and five additional strains associated with cervical cancer (HPV-31, 33, 45, 52, and 58) [22].

Although the introduction of prophylactic HPV vaccines represents a major breakthrough in cancer prevention, achieving access and widespread immunisation coverage remains challenging [23]. Many young women in sub-Saharan Africa continue to be significantly impacted by HPV infections. Persistent HR-HPV infection alone does not always result in cervical intraepithelial neoplasia (CIN) or ICC. Other factors, such as immune responses, cervicovaginal microbiota, and HIV status, play significant roles [24,25]. Women living with HIV exhibit a greater prevalence of genital HR-HPV infections compared to the general population [26], and HIV co-infection with HR-HPV is associated with an increased risk of progressing to CIN or cervical cancer, likely due to the immunosuppressive effects of HIV [27]. Furthermore, studies have indicated a potential connection between vaginal microbiota and CIN development, suggesting that a dysbiotic vaginal environment may significantly influence the onset and progression of CIN [28,29]. Vaginal dysbiosis, characterised by an overgrowth of anaerobic bacteria, promotes HPV persistence and progression by creating favourable conditions for the virus [30]. This imbalance, often leading to *Bacterial Vaginosis* (BV) [31], disrupts the local immune response, hindering the host’s ability to clear HPV infections.

A competent immune system is crucial for the clearance of viral infections [24]. Recent studies suggest a significant correlation between immune status and viral infection, with the vaginal microbiota playing a role in modulating the immune system of the female genital tract (FGT) [32,33]. Previous data indicate a correlation between changes in cytokine profiles and BV, rather than HPV infection. Few studies have examined cytokine profiles in HPV-infected and uninfected women, and uncertainties remain regarding which HPV genotypes correlate with specific cytokine profile signatures.

This study aimed to systematically review HPV research conducted in South Africa and perform a meta-analysis to uncover factors influencing the prevalence and regional distribution of HPV genotypes in South African women. Additionally, we assessed how HIV status impacts HPV infections and examined the roles of cytokine profiles and cervicovaginal microbiota in HPV prevalence and progression. By considering interactions among HPV, the microbiome, and cytokines, this study aims to inform potential strategies for cervical cancer prevention and treatment.

## 2. Methodology

### 2.1. Study Design and Search Strategy

This review was conducted in accordance with the Preferred Reporting Items for Systematic Reviews and Meta-Analyses (PRISMA) guidelines [34]. The review was not registered due to time constraints and the large number of studies being conducted in this area. Relevant articles examining HPV infection in South African women, the composition of cervicovaginal microbiota in relation to HPV, and cytokine profiles associated with HPV in this population were identified through advanced search strategies in the PubMed, Web of Science, and Scopus databases, using specified search terms. The final search for this systematic review and meta-analysis was completed on 21 September 2024. The search strategies and search terms used for each database are detailed below.

### 2.2. Search Terms

#### 2.2.1. PubMed

Search: ((((((Human papillomavirus) OR (Human Papilloma virus)) OR (Papillomaviridae)) OR (HPV)) AND (Women)) OR (Female)) AND (South Africa*) Filters: Books and Documents, Clinical Trial, Randomized Controlled Trial, English.

((((“human papillomavirus viruses”[MeSH Terms] OR (“human”[All Fields] AND “papillomavirus”[All Fields] AND “viruses”[All Fields]) OR “human papillomavirus viruses”[All Fields] OR (“human”[All Fields] AND “papillomavirus”[All Fields]) OR “human papillomavirus”[All Fields] OR (“human papillomavirus viruses”[MeSH Terms] OR (“human”[All Fields] AND “papillomavirus”[All Fields] AND “viruses”[All Fields]) OR “human papillomavirus viruses”[All Fields] OR (“human”[All Fields] AND “papilloma”[All Fields] AND “virus”[All Fields]) OR “human papilloma virus”[All Fields]) OR (“papillomaviridae”[MeSH Terms] OR “papillomaviridae”[All Fields]) OR “HPV”[All Fields]) AND (“womans”[All Fields] OR “women”[MeSH Terms] OR “women”[All Fields] OR “woman”[All Fields] OR “women s”[All Fields] OR “womens”[All Fields])) OR (“femal”[All Fields] OR “female”[MeSH Terms] OR “female”[All Fields] OR “females”[All Fields] OR “female s”[All Fields] OR “femals”[All Fields])) AND (“South”[All Fields] AND “africa*”[All Fields])) AND ((booksdocs[Filter] OR clinicaltrial[Filter] OR randomizedcontrolledtrial[Filter]) AND (english[Filter])).

#### 2.2.2. Scopus

TITLE-ABS (human AND papillomavirus) OR TITLE-ABS-KEY (hpv) OR TITLE-ABS (human AND papilloma AND virus) AND ALL (women) OR ALL (female) AND ALL (south AND africa*) AND (LIMIT-TO (LANGUAGE, “English”)).

#### 2.2.3. Web of Science

((((((TS = (Human Papillomavirus)) OR TS = (Human Papilloma virus)) OR TS = (Papillomaviridae)) OR TS = (HPV)) AND TS = (Women)) OR TS = (Female)) AND TS = (South Africa*).

Refined By: Language: English, Document types: Article.

### 2.3. Eligibility Criteria

Eligible studies for inclusion were full-text experimental research articles published in English, focusing on the following criteria: (1) HPV genotype distribution among HPV-positive South African women, regardless of HIV status; (2) research conducted in other countries that used samples from South African women; (3) all types of samples tested for HPV, regardless of infection stage or the genotyping methods used for HPV detection. The exclusion criteria were based on six key factors: (1) papers lacking methodological transparency; (2) papers that did not report on HPV DNA, HPV mRNA or utilising histological or serological methods (e.g., ELISA); (3) papers not geographically relevant (South Africa had to be included in comparative studies); (4) studies reporting on HPV in men; (5) publications in languages other than English; and (6) publications without original data (such as reviews, conference papers, guidelines, letters, and editorials).

### 2.4. Data Extraction

The records identified from the databases were downloaded in a suitable format and exported to the Endnote referencing tool. Duplicates were identified and removed using Endnote’s “Find Duplicates” feature. Titles and abstracts were then thoroughly screened, and the remaining records were assessed for eligibility by downloading and reviewing the full articles. The reviewers extracted data from the papers using Microsoft^®^ Excel 365 spreadsheets. The following variables were extracted: first author, title, year of publication, total sample size, number of HPV-positive cases, study period, study design, study location, detection methods, sample type, and province; these items are provided in the Appendix A. Any disagreements were resolved by discussing inconsistencies until a consensus was reached.

The articles were thoroughly analysed and grouped into four objectives as follows: (Objective 1) articles were included to determine the prevalence and distribution of HPV genotypes in South African women; (Objective 2) articles were included to assess the prevalence of HPV genotypes based on HIV status in South African women; (Objective 3) articles were included to examine the cervicovaginal microbiota composition and its correlation with HPV in South African women; and (Objective 4) articles were included to investigate the cytokine profiles associated with HPV in South African women (Figure 1).

### 2.5. Study Quality and Bias Assessment

Two reviewers, N.D.M. and C.K.M., independently assessed the quality of all included articles using the Joanna Briggs Institute (JBI), a Critical Review Checklist for Studies Reporting Prevalence Data [35]. Each study was categorised by its design and evaluated using a set of questions tailored to that design. Any disagreements between the reviewers were resolved through discussion and consensus. If consensus could not be reached, a third reviewer, P.P.M., was consulted for further input.

### 2.6. Statistical Analysis

The 95% confidence intervals (CIs) for the pooled prevalence estimates were analysed using the “meta” (version 8.0-1) and “metafor” (version 4.6-0) packages in RStudio for the meta-analysis. Forest plots were generated to visualise the prevalence of HPV and assess heterogeneity across studies. Heterogeneity was evaluated using the I^2^ statistic and Cochran’s Q test, with *p* < 0.1 indicating significant heterogeneity. I^2^ values above 50% were considered to reflect substantial variability among studies. Heterogeneity was categorised as follows: I^2^ values of 0%, <25%, 25–50%, and >50% indicating no heterogeneity, low heterogeneity, moderate heterogeneity, and high heterogeneity, respectively [36,37]. As a result, a random-effects model (restricted maximum-likelihood method) was applied rather than a fixed-effects model. To address the high heterogeneity, subgroup analyses were conducted based on province, sample type, and HIV status. In the province-specific subgroup analysis, studies that did not specify the province(s) from which samples were obtained, or that included participants from multiple provinces without disaggregated data, were excluded. For the subgroup meta-analysis by HIV status, studies were included only if they explicitly reported HPV prevalence according to the participants’ HIV status. Studies that either did not specify HIV status or combined HPV data from both HIV-positive and HIV-negative participants were excluded from this analysis to ensure specificity in the results. In the subgroup analysis by sample type, studies reporting HPV prevalence from two different sample types collected from the same participants were treated as separate populations. This approach allowed each sample type to be analysed independently, ensuring that the data accurately reflected each sample type and enabling more precise subgroup comparisons. Consequently, these exclusions may have affected the random-effects model, potentially altering the overall HPV prevalence estimate by approximately 1%. For sensitivity analysis both fixed-effect and random-effects models were assessed to determine the robustness of the pooled HPV prevalence estimate. The contrast in the results of both the models highlights the impact among the studies variability on the pooled prevalence. In the forest plot for subgroup analysis, the chi-square statistic (χ^2^) was used to test for differences across subgroups, with *p*-values < 0.05 indicating greater variability between them.

The extracted genotype data for HR-HPV types, LR-HPV types, and probable HR-HPV types were grouped under the HR-HPV category for analysis. When studies presented data from multiple detection methods, the method with the highest sensitivity was selected to ensure consistency across the studies included in the meta-analysis. For studies reporting HPV results at multiple follow-up time points (e.g., 6-month and 12-month follow-ups), data from the earliest time point within each period were extracted. This approach was adopted as participation typically decreases at later follow-ups, which could lead to an underestimation of HPV prevalence.

The synthesis of microbiota data included only studies reporting the composition of specific bacterial species to maintain consistency in analysis. Data on bacterial species were extracted and grouped based on their presence in HPV-positive and high-risk HPV (HR-HPV) individuals. Studies reporting microbiota data in other formats, such as the number of individuals with bacterial vaginosis (BV)-associated microbiota by HPV status, were excluded to maintain consistency. However, these data were used to support and validate the overall findings on bacterial species distribution according to HPV status.

For cytokine profile data, not all studies used the same categories. To enable comparative analysis, cytokine data were standardised into a common format. This was necessary as some studies differentiated cytokine levels by HPV-negative and HPV-positive status, while others included subgroup analyses based on HIV status or BV status alongside HPV infection. This approach ensured consistent analysis and synthesis of cytokine data across the included studies.

## 3. Results

### 3.1. Study and Population Characteristics

Our search strategy identified a total of 20,330 records from PubMed (n = 4902), Scopus (n = 4505), and Web of Science (n = 10,923). After removing 565 duplicates, 19,765 articles were included in the screening process. Of these, 19,479 articles were excluded after reviewing the titles and abstracts. Following the application of our inclusion and exclusion criteria, 120 studies were included in the meta-analysis after reviewing the full texts of 279 articles (Figure 1 and Table 1). Excluded studies and reasons for exclusion are recorded in Appendix A.

The data from the 120 included studies were conducted across five provinces in South Africa, Eastern Cape, Gauteng, KwaZulu-Natal, Limpopo, and Western Cape, spanning from 1989 to 2024. Notably, no studies were reported from the remaining four provinces of the country. In total, the studies included 83,266 women as participants (Table 1).

### 3.2. Pooled Prevalence of HPV Infection in South Africa

The overall prevalence of HPV infection in South African women, as reported in the included studies, was assessed irrespective of HIV status and sample type (Figure 2). Using the fixed-effects model, the pooled HPV prevalence was estimated at 33% (95% CI: 33–34%). However, due to the significant variability across studies, the random-effects model, which accounts for differences between studies, provides a more reliable estimate of 58% (95% CI: 52–64%). The heterogeneity among the studies was substantial, with an I^2^ of 99.1% and τ^2^ = 0.9489 (*p* < 0.001), indicating that the variation in HPV prevalence across studies was not due to chance, but likely reflects differences in study populations and methodologies. This high level of heterogeneity justifies the use of the random-effects model to accurately summarise the true burden of HPV in this population.

### 3.3. Pooled Prevalence of HPV Infection by Province

The meta-analysis revealed varying HPV prevalence rates among women across different provinces in South Africa (Appendix A). Gauteng had the highest prevalence at 64% (95% CI: 55–73%), followed by Eastern Cape at 63% (95% CI: 39–84%) and Limpopo at 59% (95% CI: 44–72%). KwaZulu-Natal and Western Cape had similar prevalence rates of 53% (95% CI: 41–65%) and 53% (95% CI: 43–62%), respectively. The random-effects model indicated significant heterogeneity across studies (I^2^ = 99.4%, τ^2^ = 0.0917, *p* < 0.001), suggesting that the observed differences in prevalence are due to genuine variations in study populations and methodologies, rather than random chance. The test for subgroup differences showed no statistically significant differences in HPV prevalence between provinces (χ^2^ = 3.48, df = 4, *p* = 0.48). This suggests that, although certain provinces show slightly higher or lower prevalence rates, the overall HPV prevalence remains high across the country.

### 3.4. Geographic Distribution of the Sample Types Studied Among South African Provinces

The geographic distribution of sample types studied across South African provinces was analysed (Figure 3). Cervical samples were the most frequently investigated, featured in 92 studies, primarily from the Western Cape (38 studies) and Gauteng (24 studies), with additional research from KwaZulu-Natal, Eastern Cape, and Limpopo. Vaginal samples were included in 12 studies, the majority of which (6 studies) came from KwaZulu-Natal, followed by studies from the Western Cape, Eastern Cape, and Gauteng. There were both cervical and oral samples (5 studies, predominantly from the Western Cape). Oral samples were collected in four studies, mainly from Gauteng and the Western Cape, while anal samples were limited to two studies from Gauteng. Less common sample types included breast tissue (1 study, Gauteng), laryngeal samples (1 study, Western Cape) and cervical and blood samples (1 study, Western Cape). Biopsy samples from genital warts were analysed in one study in Western Cape Province.

### 3.5. Pooled Prevalence of HPV Infection by Sample Type

The meta-analysis assessed HPV prevalence across various sample types, revealing significant variation (Appendix A). Using a random-effects model, the overall pooled prevalence of HPV was calculated, showing considerable heterogeneity across sample types (I^2^ = 99.4%, τ^2^ = 0.1003, *p* < 0.01). Cervical samples had the highest pooled prevalence at 60% (95% CI: 53–66%), followed by vaginal samples at 53% (95% CI: 39–67%). Anal samples showed a prevalence of 42% (95% CI: 37–47%), while oral samples had a lower prevalence of 25% (95% CI: 9–47%). Genital wart biopsy, blood, and breast tissue samples displayed relatively lower prevalence rates, ranging from 17% to 37%. Laryngeal samples had the lowest prevalence at 0.12%. The test for subgroup differences revealed significant variability between sample types (χ^2^ = 514.15, df = 7, *p* < 0.01), highlighting that the type of biological sample plays a critical role in determining HPV prevalence. The observed heterogeneity indicates that these differences in HPV prevalence are due to genuine variations based on sample type, rather than random variation.

### 3.6. Pooled Prevalence of HPV Infection by HIV Status

The meta-analysis evaluated the prevalence of HPV based on HIV status among women (Appendix A). Using a random-effects model, the HPV prevalence was estimated, revealing significant heterogeneity across studies and sample types (I^2^ = 99%, τ^2^ = 0.0821, *p* < 0.001). HIV-positive women had the highest prevalence at 63% (95% CI: 55–71%), while HIV-negative women had the lowest prevalence at 39% (95% CI: 29–50%). The test for subgroup differences showed significant variability between sample types (χ^2^ = 11.89, df = 1, *p* < 0.01), emphasising the differences in HPV prevalence based on HIV status among women.

### 3.7. Overall HR-HPV Genotype Prevalence

The prevalence of high-risk (HR) HPV genotypes across detection periods from 1989 to 2024 was summarised in 70 studies (Figure 4). The prevalence rates of the 22 HR-HPV genotypes reported ranged from 0.1% to 7.6%. The most prevalent genotype was HPV-16, detected in 65 studies from 1989 to 2024, with a prevalence of 7.50%. This was followed by HPV-35 (4.11%) and HPV-18 (3.87%). The widespread detection of HPV-16 and HPV-35 highlights their significant presence in the population, making them the most prevalent HR-HPV genotypes in this analysis. Other notable high-risk genotypes include HPV-52 (3.42%), HPV-58 (2.98%), and HPV-45 (2.95%). Lower-prevalence types, such as HPV-26 (0.19%) and HPV-67 (0.16%), were also observed. These results provide a comprehensive overview of the distribution of HR-HPV genotypes over time, with the detection periods indicating when each genotype was first and most recently identified.

### 3.8. The Trends in the Detection of the Most Prevalent HR-HPV Genotypes Studied (16, 18, 35, and 52) from 2005 to 2024

Over the observed period from 2005 to 2024, the prevalence of high-risk (HR) HPV genotypes 16, 18, 35, and 52 showed significant variability (Figure 5). These findings highlight the persistence of HPV-16 as the most prevalent type, with fluctuations in the detection rates of HPV-18, 35, and 52 over time. HPV-16 consistently exhibited the highest detection rates, maintaining a strong presence throughout the years from 2005 to 2024. Similarly, HPV-18 had high detection rates during this period, while HPV-35 showed sporadic detections and HPV-52 was only detected from 2006 onwards. Notably, from 2015, HPV-35 had higher detection rates than HPV-18, with both genotypes showing varying levels of detection, each peaking in different periods but generally remaining within the 0–20% range. HPV-52, among the other genotypes, had lower detection rates but remained relatively consistent throughout the observed period. The peaks in 2015 for all four HPV types may indicate a period of increasing detection or prevalence, possibly reflecting improvements in testing or diagnostic methods. However, there were years, such as 2013 and 2020, in which one or more HPV genotypes were not detected, suggesting potential changes in diagnostic capabilities or shifts in research focus.

### 3.9. HR-HPV Genotype Prevalence in Female Genital Tract and Upper Respiratory Tract

The prevalence of high-risk (HR) HPV genotypes in female genital tract (FGT) samples (including cervical and vaginal samples) among South African women was assessed from 1989 to 2024 across 65 studies (Appendix A). In contrast, HR-HPV was detected in upper respiratory tract (URT) samples (including oral and laryngeal samples) from 2002 to 2020 across three studies (Appendix A). HPV-16 was the most common genotype found in both sample types, with a prevalence of 6.61% in FGT samples and 7.24% in URT samples. This was followed by HPV-35 (3.65%) and HPV-18 (3.48%) in FGT samples. However, in URT samples, HPV-18 (2.41%) was the second most prevalent genotype, while HPV-35 was not detected. Other notable genotypes included HPV-52 (3.11%) and HPV-58 (2.98%) in FGT samples, and HPV-33 (1.61%) and HPV-68 (0.54%) in URT samples. Similar trends were observed across both sample types, with low prevalence of HPV-70 and HPV-83. These genotypes were the least prevalent in URT samples, both at 0.27%. In FGT samples, the lowest prevalence was observed for HPV-67 and HPV-26, with both showing a prevalence of less than 0.2%. These data reflect the distribution of HR-HPV genotypes in FGT and URT samples collected over time.

### 3.10. Overall LR-HPV Genotype Prevalence

The prevalence of low-risk (LR) HPV genotypes across the included studies, covering detection periods from 1991 to 2023 were analysed (Figure 6). Overall, the prevalence of LR-HPV genotypes ranged from 0.1% to 6.0% across all 28 studies. The most prevalent genotype was HPV-62, with a prevalence of 5.80%, followed by HPV-6 (5.47%), HPV-84 (5.26%), and HPV-11 (4.73%). This widespread detection highlights the significant presence of HPV-62 and HPV-84 in the population. Other notable LR genotypes included HPV-61 (4.35%), and HPV-81 and HPV-54, both with prevalences of less than 3.1%. The lowest prevalence was observed for HPV-43 and HPV-74, both with a prevalence of 0.17%.

### 3.11. LR-HPV Genotype Prevalence in Female Genital Tract and Upper Respiratory Tract Sample Types

The prevalence of low-risk (LR) HPV genotypes in the female genital tract (FGT) and upper respiratory tract (URT) regions, as observed from 1991 to 2023, showed considerable variation (Appendix A). Across all 18 LR-HPV genotypes, prevalence rates ranged from 0.1% to 6.0% in FGT samples and from 0.4% to 19% in URT samples. In FGT samples, HPV-62 was the most prevalent genotype at 5.84%, followed closely by HPV-6—a globally common type—at 5.47%, and HPV-84 at 5.30%. HPV-11, another globally common type, showed a prevalence of 4.4%. Other notable LR genotypes in FGT samples included HPV-61 (4.38%), HPV-81 (3.17%), and HPV-54 (3.10%).

In comparison, HPV-11 was the most prevalent genotype in URT samples, with a prevalence of 18.38%, followed by HPV-6 and HPV-72, both at 2.70%. Less prevalent genotypes in URT samples included HPV-61, HPV-71, HPV-81, HPV-84, and HPV-89, each with a prevalence of 0.54%. These data provide a comprehensive overview of the distribution of LR-HPV types in FGT and URT samples over time.

### 3.12. Prevalence of HR and LR-HPV Genotypes Stratified by HIV Status

The distribution of high-risk (HR) HPV genotypes was analysed in both HIV-positive and HIV-negative women (Figure 7). Among HIV-negative women, HPV-16 was the most prevalent genotype, accounting for 3.92% of infections, followed by HPV-35 (2.56%) and HPV-18 (1.71%). The least prevalent genotypes in this group were HPV-26, HPV-69, and HPV-83. In contrast, HIV-positive women showed a higher prevalence of HR-HPV genotypes, with HPV-16 also being the most common, detected in 11.01% of cases. Other frequently detected genotypes in this group included HPV-18 (6.41%), HPV-35 (6.15%), and HPV-45 (5.89%). Low-prevalence HR-HPV genotypes in HIV-positive women included HPV-67 and HPV-70, with detection rates of 0.02% and 0.69%, respectively, while these were undetected in HIV-negative women.

The distribution of low-risk (LR) HPV genotypes also differed significantly between HIV-positive and HIV-negative South African women (Figure 8). In HIV-positive women, the most prevalent LR-HPV genotypes were HPV-62 (9.66%) and HPV-11 (8.97%), followed by HPV-61 (8.45%) and HPV-83 (5.17%). Genotypes with low prevalence rates in this group included HPV-42 (0.34%), HPV-40 (0.52%), and HPV-71 (0.52%). Among HIV-negative women, the most prevalent LR-HPV genotypes were HPV-11 (5.21%), HPV-6 (4.17%), and HPV-72 (1.88%), with other notable genotypes including HPV-40, HPV-81, and HPV-84, each at 1.25%. Overall, LR-HPV genotypes, particularly HPV-6 and HPV-11, were more prevalent in HIV-positive women compared to HIV-negative women.

### 3.13. Association Between Cervicovaginal Microbiota and HPV Infection

The *network analysis* provides a comparative summary of all bacterial species found to be either significantly or non-significantly correlated with various HPV infections from two key studies included in this review [124,145] (Figure 9). Additional studies [118,131,134,140,141] contributed supplementary findings; however, direct quantitative comparisons were constrained by inconsistencies in data reporting and variation in the bacterial species studied. Nonetheless, these studies consistently indicated that individuals with HPV were more likely to exhibit a microbiome highly associated with *BV* compared to HPV-negative individuals.

In the HPV-negative/*LR-HPV*-positive category, *Lactobacillus crispatus* was significantly correlated, suggesting a microbiome dominated by lactobacilli, which is typically considered protective. Other species showing significant associations included *Corynebacterium tuberculostearicum*, *Corynebacterium amycolatum/Corynebacterium lactis*, *Pasteurellales*, *Alphaproteobacteria*, *Gammaproteobacteria*, *Pseudomonadales*, *Phyllobacterium*, and *Rhizobiales*.

Bacterial species positively correlated with HPV positivity included a mixture of *BV*-associated bacteria. Significant correlations with HPV-positive infection were observed for *Aerococcus christensenii*, *Prevotella*, *Atopobium vaginae*, *Gardnerella vaginalis*, *Coriobacteriaceae*, *Dialister micraerophilus*, *Dialister succinatiphilus/D. propionifaciens*, *Megasphaera*, *Fannyhessea vaginae*, *Gemella asaccharolytica*, *Peptoniphilus lacrimalis*, and *Parvimonas micra*. Notably, *Prevotella*, *Megasphaera*, and *Gardnerella vaginalis* demonstrated strong associations with *BV*.

In HR-HPV-positive women, a distinct microbial composition was observed, including Bifidobacteriales, Bifidobacteriaceae, Aerococcus, Aerococcaceae, Coriobacteriales, Coriobacteriaceae, Fusobacteriales, Fusobacteria, Actinobacteria, Gardnerella, Gardnerella vaginalis, Atopobium, Atopobium vaginae, Leptotrichiaceae, and Sneathia. Among these, Coriobacteriales, Actinobacteria, Fusobacteria, Fusobacteriales, Gardnerella, Gardnerella vaginalis, Atopobium, and Atopobium vaginae were significantly associated with BV, indicating a distinct microbiome profile in HR-HPV cases characterised by a higher prevalence of BV-associated bacteria.

### 3.14. HPV Infection Association with Cervicovaginal Cytokine Profile

The relationship between HPV infection and cervicovaginal cytokine levels was analysed using data from three key studies [105,123,145]. These studies collectively measured 55 cytokines, offering a detailed overview of immune responses in the cervicovaginal environment across different infection statuses. The Venn diagrams illustrate cytokine profiles associated with HPV positivity, HIV positivity, and BV presence in cervicovaginal samples, highlighting both shared and unique cytokine signatures for each condition. Cytokines significantly associated with each category, adjusted for multiple comparisons, are marked with an asterisk in boxes adjacent to each category in the Venn diagram (Figure 10).

In HPV-negative individuals, IL-10, IL-15, and MCP-3 were detected at lower levels, although these differences did not reach statistical significance. Conversely, HPV-positive individuals displayed elevated levels of IL-1β, IL-17F, IL-25, IL-33, and TNF-α, which overlapped with cytokine profiles associated with BV. However, following adjustments for multiple comparisons, these cytokines (IL-1β, IL-17F, IL-25, IL-33, and TNF-α) were significantly associated only with BV. In the HIV-positive group, there was an overlap in elevated levels of IP-10, IL-8, and IL-6, shared with the HPV-positive category. Additionally, cytokines IL-1α, IL-6, IP-10, G-CSF, and MCP-1 were significantly associated with HIV positivity. Women infected with HPV had notably higher mucosal concentrations of MIP-1α and MIP-1β, suggesting a distinct cytokine profile associated with HPV positivity.

Happel et al. [145] found that cervicovaginal cytokine levels varied by HPV genotype. High-risk HPV types 35, 39, and 68 were associated with elevated levels of cytokines such as IL-17A, IL-17F, IL-25, TNF-α, and IFN-γ, while HPV-30 was linked to reduced cytokine levels. HPV-35 infection remained significantly associated with elevated levels of six cytokines, including IL-1β and TNF-α, even after controlling for multiple comparisons and BV status.

### 3.15. Publication Bias Analysis

A funnel plot was generated to assess potential publication bias among the included studies (Appendix A). The plot reveals some asymmetry, especially in the lower half, where studies with smaller sample sizes and higher standard errors are scattered unevenly. This may indicate possible publication bias, as studies with lower HPV prevalence and smaller sample sizes appear under-represented in the analysis. Alternatively, the asymmetry could be influenced by study heterogeneity, including variations in populations, methodologies, or HPV genotype prevalence. Egger’s test results further support this visual finding, with a statistically significant z-score of 1.9831 and a *p*-value of 0.0474, suggesting the presence of publication bias in the analysed studies.

### 3.16. Risk of Bias Assessment

The studies were grouped based on their design, and the risk of bias varied across these categories. Among cross-sectional and case-control studies (n = 64), 56% were assessed as having a low risk of bias, 41% as moderate risk, and 3% as high risk. For case-control studies specifically (n = 9), 56% were rated as low risk, 33% as moderate risk, and 11% as high risk. Cohort studies (n = 41) showed 42% with low risk, 51% with moderate risk, and 7% with high risk. In contrast, case series (n = 6) had the highest proportion of high-risk studies, with 50% rated as high risk, 33% as moderate risk, and only 17% as low risk.

Overall, across all included studies (n = 120), 49% (59/120) were rated as low risk, 42% (50/120) as moderate risk, and 9% (11/120) as high risk. This indicates that while most studies were of acceptable quality, nearly half had a moderate to high risk of bias, which could impact the reliability of the findings. Detailed risk of bias scores are provided in the Appendix A.

## 4. Discussion

This systematic review and meta-analysis provide critical insights into HPV prevalence, genotype distribution, and its associations with cervicovaginal microbiota and cytokine profiles in South African women, particularly within the context of HIV status.

Our meta-analysis findings reveal a pooled HPV prevalence of 58% (95% CI: 52–64%) among South African women, underscoring a substantial burden of infection. These results align with findings by Ogembo et al. [155], who reported an HPV prevalence of 57.3% in Southern Africa, higher than other African regions, highlighting a significant infection rate in the region. Provincial prevalence rates ranged from 53% in KwaZulu-Natal and the Western Cape to 64% in Gauteng, with regional gaps in research (e.g., Mpumalanga, Northwest, Northern Cape, and Free State) limiting a full understanding of national patterns.

One of the most significant findings is the prevalence of HPV-35 (4.11%), which is disproportionately high in the African population compared to global data, where HPV-16 and -18 typically dominate. The increasing prevalence of HPV-35 among African populations, also reported by Bouassa et al. [156] and Tchouaket et al. [157], underscores a critical limitation in current HPV vaccine coverage, which primarily targets HPV-16 and -18. Expanding vaccine formulations to cover additional high-risk genotypes like HPV-35 could significantly enhance the effectiveness of HPV vaccination programmes across Africa, potentially reducing cervical cancer rates that remain disproportionately high in the region. These results call for global health organisations to consider more inclusive vaccine-development strategies that address diverse genotype distributions across populations. Among LR-HPV types, HPV-62 (5.80%) and HPV-84 (5.26%) were the most common, further reflecting the diversity of circulating HPV genotypes in South Africa.

Our study also examined HPV prevalence across various sample types, aiming for a broad view of the viral landscape. Cervical samples showed the highest prevalence (60%), followed by vaginal (53%), anal (42%), and oral samples (25%). The high prevalence in FGT samples, as compared to the URT, may reflect biological differences in HPV persistence and suggests the benefit of concurrent sampling from cervical and vaginal sites to capture the full scope of infection within the FGT. The lower prevalence and diversity of HPV in non-genital samples may also be due to reduced HPV viral loads or lower test sensitivity at these sites, as noted in previous studies [158]. This highlights the need for future HPV genotyping research to better understand viral dynamics in non-genital locations, which contribute to broader transmission and potential oncogenesis.

The analysis also confirmed a higher pooled HPV prevalence (63%) in HIV-positive women compared to HIV-negative women (39%), consistent with global research linking HIV-associated immunosuppression to higher HPV acquisition and persistence rates [123,126,159], yet the magnitude observed here calls for heightened integration of HPV and HIV prevention strategies in African contexts. Enhanced screening and treatment programs that address both HIV and HPV could improve outcomes for co-infected individuals and alleviate some of the public health burden linked to cervical cancer in Africa.

Given the high prevalence of HIV and HPV co-infection in Africa, understanding the role of the vaginal microbiota in modulating immune responses to these infections is essential for developing targeted screening and treatment strategies. The vaginal microbiota contains array of microorganisms that are crucial for both human health and the development of various diseases [29]. Our findings reveal a shift from *Lactobacillus*-dominance to a more diverse, BV-associated microbiome in HPV-positive and HR-HPV-positive women, a pattern that echoes findings in other global populations. This suggests that a non-*Lactobacillus*-dominated environment may facilitate HPV persistence and potentially influence disease outcomes; however, the bidirectional influence of microbiota and disease progression requires further study. BV-associated species such as *Atopobium*, *Aerococcus*, *Gardnerella*, *Sneathia*, and *Pseudomonas* are potential biomarkers for HPV infections [123], warranting further exploration as targets for therapeutic or probiotic interventions.

Distinct cytokine profiles linked to HPV, HIV, and BV status also highlight immune mechanisms that may influence infection persistence or clearance. Consistent with research on chronic HIV infection and elevated genital inflammatory cytokines [160,161]. HIV-positive women exhibited higher levels of IL-1α, MCP-1, and G-CSF, while IP-10, IL-6, and IL-8 were elevated in both HPV-positive and HIV-positive groups. The overlap of specific cytokines in HPV- and HIV-positive individuals suggests a shared inflammatory environment that could exacerbate HPV persistence in co-infected individuals. In contrast with other studies [162,163,164], HPV-negative individuals showed lower levels of cytokines like IL-10 and IL-15, possibly reflecting a baseline immune environment with reduced inflammation. Notably, associations between BV-related bacteria and cytokines IL-1β and TNF-α in HPV-35-positive individuals highlight complex interactions between the microbiome and immune responses in HPV infection. In the broader context, these findings call attention to the potential for cytokine-based biomarkers to guide treatment strategies and enable early identification of women at risk for persistent infection or disease progression. Such biomarkers would be especially valuable in settings with limited access to advanced diagnostics.

## 5. Limitations

Our review underscores the need for greater regional diversity in HPV research across South Africa, as many provinces remain understudied, leading to gaps in understanding HPV distribution at a national level. Future research should also focus on HPV prevalence in non-genital sites to improve understanding of broader transmission dynamics, especially considering oral HPV’s emerging role in global HPV transmission patterns. Most of the included studies had a low to moderate risk of bias, indicating generally acceptable methodological quality. However, the observed risk of bias could have introduced variability, potentially affecting the robustness of the meta-analysis. This is supported by the asymmetry detected in the publication bias. Finally, addressing inconsistencies in microbiome and cytokine data reporting would facilitate more robust meta-analyses in future studies, enabling deeper insights into the biological interactions between HPV, the microbiome, and immune responses.

## 6. Conclusions

In summary, this systematic review and meta-analysis highlights South Africa’s high burden of HPV and its unique genotype distribution, with broader implications for vaccine development, screening practices, and integrative HPV- and HIV-prevention strategies. These findings contribute meaningfully to understanding HPV’s public health impact in Southern Africa, where prevalence rates are notably high. When considered alongside studies from other regions of Africa, our results highlight a pressing need for tailored public health strategies, as the prevalence in Southern Africa exceeds that reported in other parts of the continent. This disparity reinforces the urgent need for region-specific approaches to HPV prevention, screening, and treatment. Additionally, these findings advocate for more inclusive vaccine coverage and underscore the importance of addressing microbial and immune factors in HPV research, particularly in African contexts. Addressing geographic and sample-type disparities in South African HPV research is essential for a more comprehensive understanding of infection patterns.

## Figures and Tables

**Figure 1 viruses-16-01893-f001:**
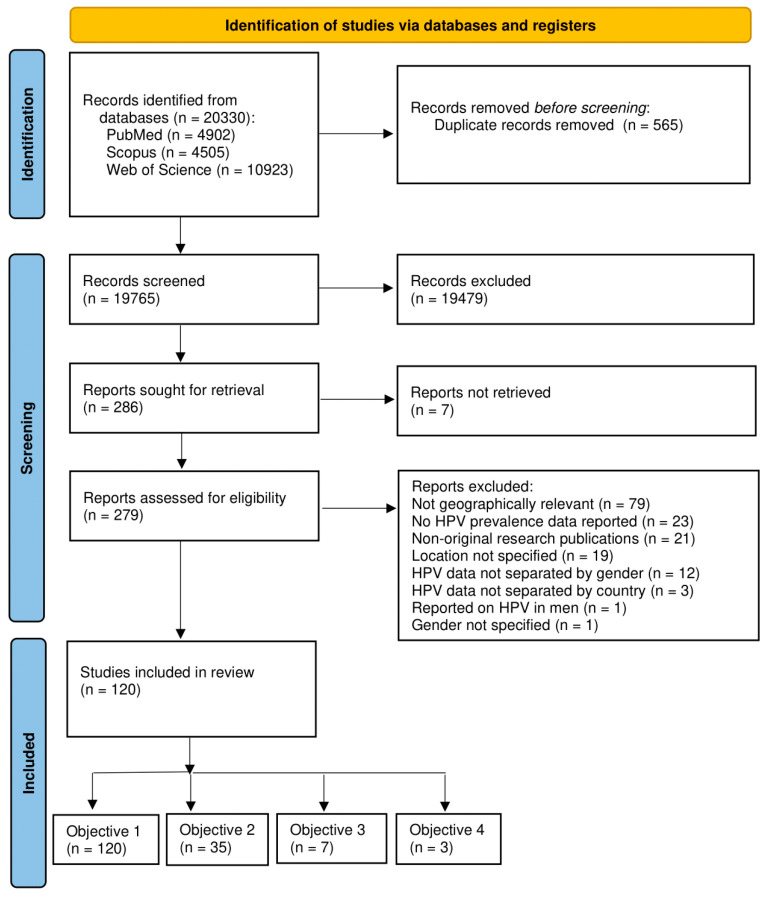
PRISMA (Preferred Reporting Items for Systematic Reviews and Meta-Analyses) flow diagram for studies on the prevalence and distribution of HPV types and their interaction with cervicovaginal microbiota and cytokine profile among women living in South Africa.

**Figure 2 viruses-16-01893-f002:**
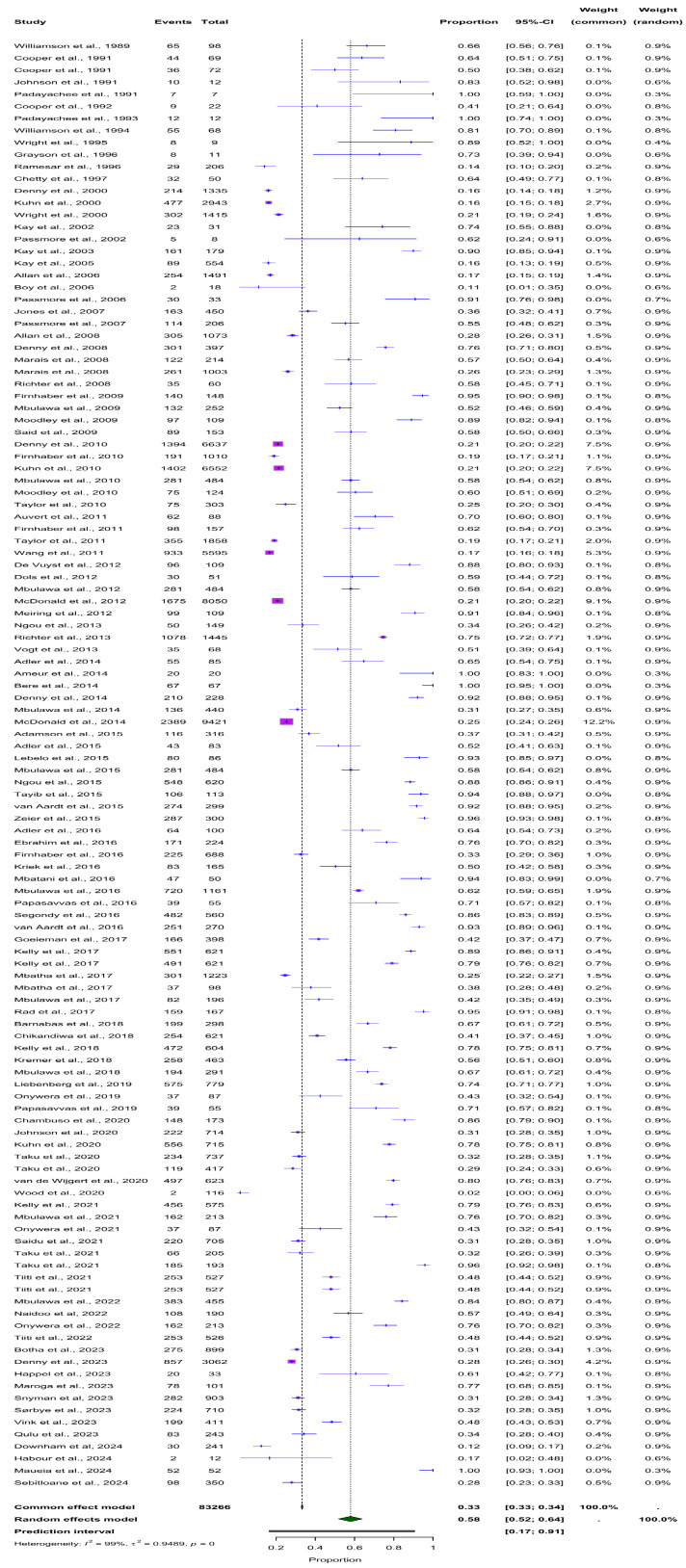
Forest plot showing the overall prevalence of HPV in South African women based on various studies. The random effects model estimated a pooled prevalence of 58% (95% CI: 52–64%). Significant heterogeneity was observed among the included studies (I^2^ = 99.1%, τ^2^ = 0.9489, *p* < 0.001).

**Figure 3 viruses-16-01893-f003:**
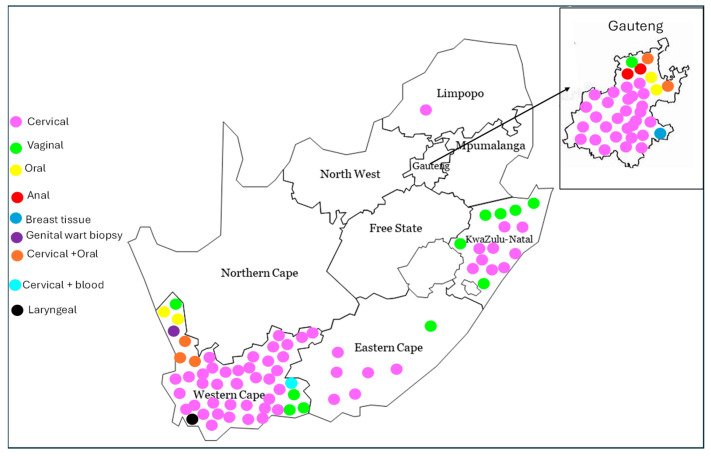
Geographic distribution of sample types used in HPV studies across South African provinces. The map illustrates the geographic distribution of various sample types used in HPV studies across South Africa. Each dot represents an individual study, and the colour of the dots indicates the sample type: pink for cervical, green for vaginal, yellow for oral, red for anal, blue for breast tissue, brown for cervical and oral, black for laryngeal, turquoise for cervical and blood, and purple for biopsy. The position of each dot within a province does not necessarily reflect the exact location of the study within that province. The map highlights key regions, such as Gauteng and the Western Cape, where HPV research has been most prominent. The Gauteng province, located in the top-right corner, is zoomed out to ensure all sample dots are visible.

**Figure 4 viruses-16-01893-f004:**
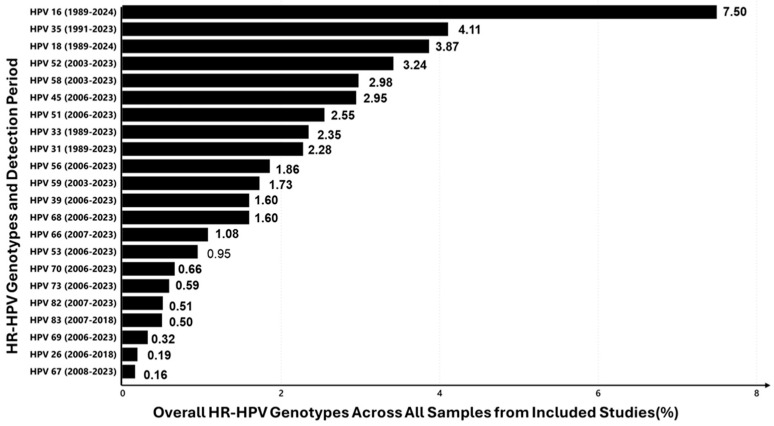
Overall prevalence (%) of HR-HPV genotypes detected in studies conducted from 1989 to 2024.

**Figure 5 viruses-16-01893-f005:**
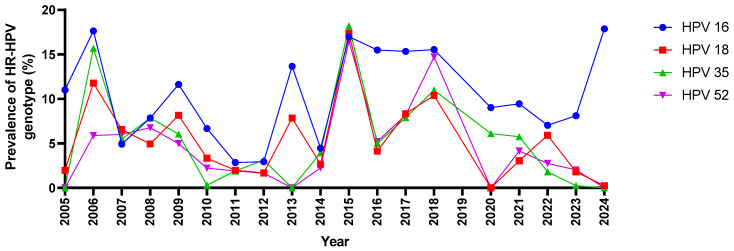
Detection trends of HPV types 16, 18, 35, and 52 from 2005 to 2024. The graphs show the percentage detection of each type over time from all the studies included. Peaks in detection for HPV-16 and HPV-18 are observed from 2005 to 2024, while HPV-35 and 52 show variable detection trends across the years.

**Figure 6 viruses-16-01893-f006:**
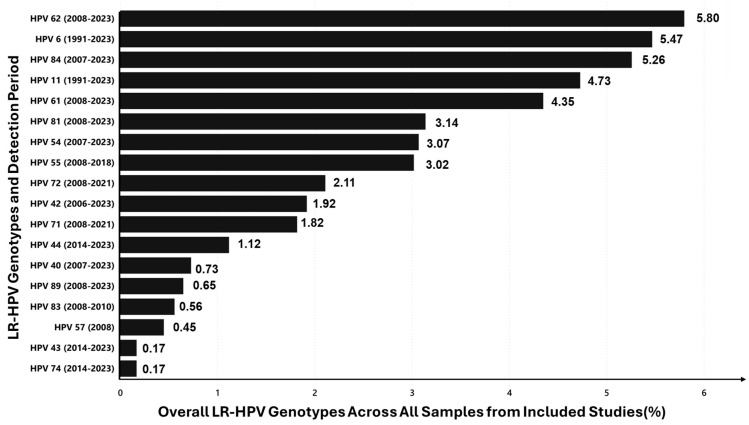
Overall prevalence (%) of LR-HPV genotypes detected in studies conducted from 1996 to 2023.

**Figure 7 viruses-16-01893-f007:**
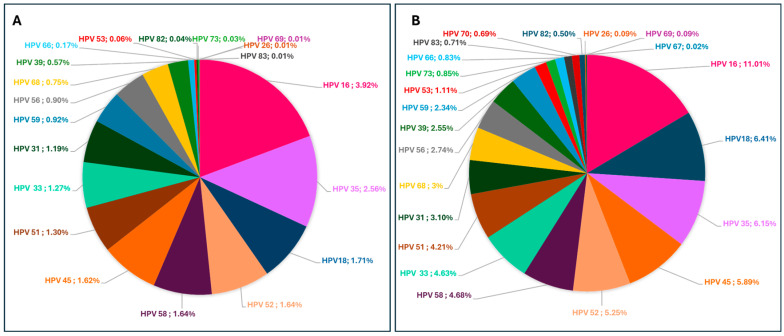
Distribution and prevalence of HR-HPV genotypes among HIV-negative (**A**) and HIV-positive (**B**) women. The pie charts represent the percentage prevalence of each HR-HPV genotype within each group.

**Figure 8 viruses-16-01893-f008:**
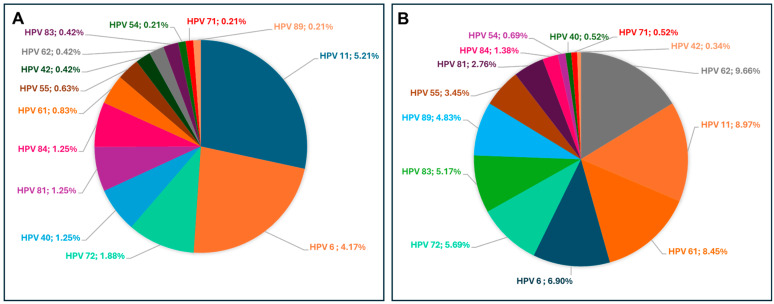
Distribution of LR-HPV genotypes among HIV-negative (**A**) and HIV-positive (**B**) women. The pie charts represent the percentage prevalence of each LR-HPV genotype within each group.

**Figure 9 viruses-16-01893-f009:**
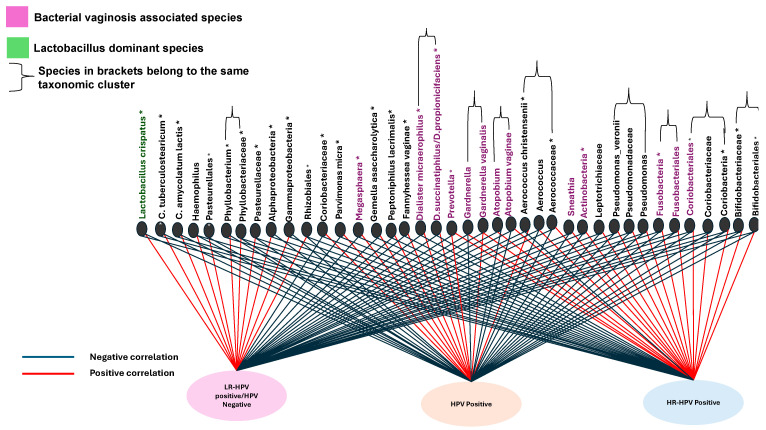
Network analysis of cervicovaginal microbiota composition correlated with HPV infection status. Negative correlations are displayed in blue, while positive correlations are in red. Bacterial species associated with bacterial vaginosis (BV) are shown in purple, *Lactobacillus*-dominated species are in green, and those bacterial species with association not clearly indicated are shown in black. Taxonomic clusters indicate groups of taxa sharing the same hierarchical classification levels, denoting biological relatedness. Bacterial taxa significantly associated with each HPV category are marked with an asterisk above the species name.

**Figure 10 viruses-16-01893-f010:**
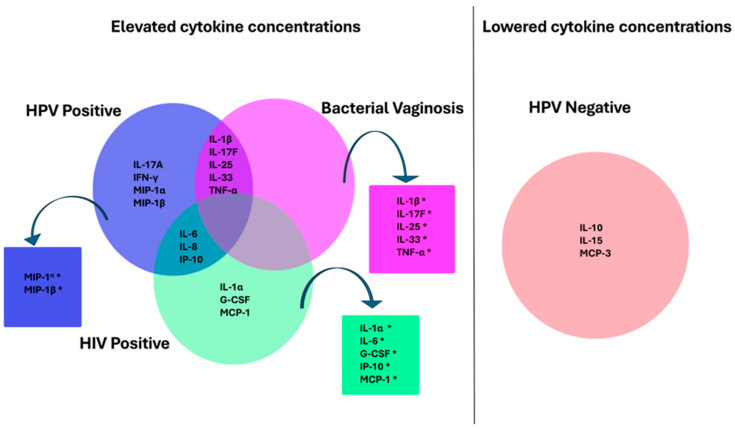
Venn diagrams illustrating elevated cytokine concentrations in women with HPV infection, HIV infection, and bacterial vaginosis (BV) and lowered cytokines in HPV-negative women. The cytokines significantly associated with each category are indicated by the asterisk.

**Table 1 viruses-16-01893-t001:** Characteristics of the 120 included studies.

Author	Year	Province	Sample Period	Sample Type	HPV Detection Method	Setting
Williamson et al. [38]	1989	Western Cape	^#^ N/A	Cervical	^ SBH	Clinic
Cooper et al. [39]	1991	KwaZulu-Natal	1998 to 1990	Cervical	^ð^ NISH	Hospital
Cooper et al. [40]	1991	KwaZulu-Natal	N/A	Cervical	NISH	Hospital
Johnson et al. [41]	1991	N/A	N/A	Cervical	SBH	N/A
Padayachee et al. [42]	1991	N/A	N/A	Oral	^$^ ISH	N/A
Cooper et al. [43]	1992	KwaZulu-Natal	1988	Cervical	NISH	Hospital
Padayachee et al. [44]	1993	Western Cape	1982 to 1988	Laryngeal	ISH	Hospital
Williamson et al. [45]	1994	Western Cape	N/A	Cervical	SBH	Clinic
Wright et al. [46]	1995	Gauteng	1991 to 1993	Vaginal	NISH	Clinic
Grayson et al. [47]	1996	N/A	1974 to 1995	Cervical	NISH	N/A
Ramesar et al. [48]	1996	Western Cape	N/A	Cervical	SBH	Clinic and private practices
Chetty et al. [49]	1997	N/A	N/A	Cervical	* PCR	N/A
Denny et al. [50]	2000	Western Cape	1996	Cervical	^α^ HCII	Clinic
Kuhn et al. [51]	2000	Western Cape	N/A	Cervical	HCII	Clinic
Wright et al. [52]	2000	Western Cape	1998 to 1999	Vaginal	HCII	Clinic
Kay et al. [53]	2002	Western Cape	1997	Oral	Nested PCR	Clinic
Passmore et al. [54]	2002	Western Cape	N/A	Cervical	PCR	Clinic
Kay et al. [55]	2003	Western Cape	1993 to 1997	Cervical	^&^ RFLP	Hospital
Kay et al. [56]	2005	Western Cape	1998 to 2001	Cervical and blood	Nested PCR	Urban and peri-urban areas
Allan et al. [57]	2006	Western Cape	1998 to 2001	Cervical	HCII	Hospitals and community health services
Boy et al. [58]	2006	Gauteng	1998 to 2003	Oral	^€^ RT-PCR	Departments of Anatomical and Oral Pathology
Passmore et al. [59]	2006	Western Cape	N/A	Cervical	^∞^ RLA	Clinic
Jones et al. [60]	2007	Western Cape	2002	Cervical	HCII	Community Health centre
Passmore et al. [61]	2007	Western Cape	N/A	Cervical and oral	HCII	Clinic
Allan et al. [62]	2008	Western Cape	1999 to 2001	Cervical	RLA	Hospitals and community health services
Denny et al. [63]	2008	KwaZulu-Natal	2002 to 2003	Cervical	HCII	Clinics
Marais et al. [64]	2008	Western Cape	N/A	Cervical and oral	RLA	Clinic
Marais et al. [65]	2008	Western Cape	N/A	Cervical	^°^ RLB	Clinic
Richter et al. [66]	2008	Gauteng	N/A	Cervical and oral	RLA	Hospitals and clinics
Firnhaber et al. [67]	2009	Gauteng	N/A	Cervical	RLA	Clinic
Mbulawa et al. [68]	2009	Western Cape	2006 to 2009	Cervical	RLA	Investigations of genital HPV transmission
Moodley et al. [69]	2009	Western Cape	2007	Cervical	RLA	Clinic
Said et al. [70]	2009	Gauteng	N/A	Cervical	RLA	Participating in a phase III microbicide study
Denny et al. [71]	2010	Western Cape	2000 to 2002	Cervical	HCII	community outreach
Firnhaber et al. [72]	2010	Gauteng	N/A	Cervical	RLA	Clinic
Kuhn et al. [73]	2010	Western Cape	2000 to 2002	Cervical	HCII	Clinics
Mbulawa et al. [74]	2010	Western Cape	2006 and 2009	Cervical	RLA	Investigations of genital HPV transmission
Moodley et al. [75]	2010	KwaZulu-Natal	N/A	Cervical	RLA	Clinics
Taylor et al. [76]	2010	Western Cape	2000 to 2002	Cervical	HCII	Community education and outreach activities
Auvert et al. [77]	2011	KwaZulu-Natal	1996 to 2000	Vaginal	RLA	Multicentre
Firnhaber et al. [78]	2011	N/A	N/A	Cervical	^¥^ HPV-4cLIA	Clinics
Taylor et al. [79]	2011	Western Cape	2000 to 2002	Cervical	HCII	Clinical trial
Wang et al. [80]	2011	Western Cape	2000 to 2002	Cervical	RLA and HCII	Cervical cancer prevention trial
De Vuyst et al. [81]	2012	KwaZulu-Natal	2007 to 2009	Cervical	^Δ^ EIA	Hospital
Dols et al. [82]	2012	Limpopo	2008	Cervical	^ʓ^ SPF10 PCR-DEIA-LiPA25	Centre
Mbulawa et al. [83]	2012	Western Cape	2006 to 2009	Cervical	RLA	Centre
McDonald et al. [84]	2012	Western Cape	1998 to 1999	Cervical	HCII	Clinics
Meiring et al. [85]	2012	Western Cape	N/A	Cervical	Illumina sequencing	Clinic
Bere et al. [86]	2013	Western Cape	N/A	Genital wart	HPV genotyping	Clinic
Ngou et al. [87]	2013	Gauteng	2011 to 2012	Cervical	HCII	Clinic
Richter et al. [88]	2013	Gauteng	2009 to 2011	Cervical	RealTime	Clinics
Vogt et al. [89]	2013	Gauteng	2011	Cervical and oral	^©^ IBH	Centre
Adler et al. [90]	2014	Western Cape	2012 to 2014	Vaginal	RLA	Centre
Ameur et al. [91]	2014	Western Cape	2006 to 2009	Cervical	RLA	Centre
Denny et al. [92]	2014	N/A	2007 to 2010	Cervical	SPF10 PCR-DEIA-LiPA25	N/A
Mbulawa et al. [12]	2014	Western Cape	2006 to 2009	Cervical and oral	RLA	Centre
McDonald et al. [93]	2014	Western Cape	2000 to 2002	Cervical	Hybrid Capture II test	Three clinical sites
Adamson et al. [94]	2015	Gauteng	2014	Cervical	Aptima HPV assay	Clinic
Adler et al. [95]	2015	Western Cape	2012 to 2014	Vaginal	RLA	Centre
Lebelo et al. [96]	2015	Gauteng	2008 to 2009	Cervical	TaqMan-based quantitative qPCR	Hospital
Mbulawa et al. [97]	2015	Western Cape	2006 to 2009	Cervical	RLA	Centre
Ngou et al. [98]	2015	Gauteng	2011 to 2012	Cervical	^A^ INNO-LiPA	Clinic
Tayib et al. [99]	2015	Western Cape	2010	Cervical	RLA	Clinic
van Aardt et al. [100]	2015	Gauteng	2003 to 2011	Cervical	RLA	Referred for staging and treatment of histologically confirmed invasive cervical cancer
Zeier et al. [101]	2015	Western Cape	2009 to2011	Cervical	RLA	^≠^ cART naïve HIV infected women
Adler et al. [102]	2016	Western Cape	October 2013 & March 2015	Vaginal	RLA	Youth community centre
Ebrahim et al. [103]	2016	KwaZulu-Natal	2004 to 2007	Vaginal	RLA	Primary health care Clinics
Firnhaber et al. [104]	2016	Gauteng	March 2010 to August 2013	Cervical	HCII	HIV treatment clinic
Kriek et al. [105]	2016	Western Cape	N/A	Cervical	RLA	Centre
Mbatani et al. [106]	2016	Western Cape	N/A	Cervical	RLA	Women diagnosed with invasive cervical cancer
Mbulawa et al. [107]	2016	Gauteng	2009 to 2011	Cervical	GeneXpert HPV	Clinic
Papasavvas et al. [108]	2016	Gauteng	N/A	Cervical	RLA	Hospital
Segondy et al. [109]	2016	Gauteng	2011 and 2012	Cervical	INNO-LiPA	Centres and clinics
van Aardt et al. [110]	2016	Gauteng	2010 to 2013	Cervical	RLA	Public healthcare facilities
Goeieman et al. [111]	2017	Gauteng	N/A	Anal	HCII	Hospital
Kelly et al. [112]	2017	Gauteng	2011 to 2012	Cervical	INNO-LiPA	HIV treatment centres
Kelly et al. [113]	2017	Gauteng	2011 to 2012	Cervical	INNO-LiPA	HIV treatment centres
Mbatha et al. [114]	2017	KwaZulu-Natal	2010 and 2013	Vaginal	EIA	Schools in two rural areas
Mbatha et al. [115]	2017	KwaZulu-Natal	2010 and 2013	Vaginal	EIA	Clinical study
Mbulawa et al. [116]	2017	Gauteng	2012 to 2014	Anal	HCII	Hospital
Rad et al. [117]	2017	Gauteng	2008 to 2011	Cervical	RLB	Institute of Community Medicine (ISM) and Department of Clinical Pathology
Barnabas et al. [118]	2018	Gauteng and Western Cape	N/A	Cervical	RLA	N/A
Chikandiwa et al. [119]	2018	Gauteng	2011 and October 2012	Cervical	INNO-LiPA	Centres and communities
Kelly et al. [120]	2018	Gauteng	2011 to 2012	Cervical	INNO-LiPA	Centres and communities
Kremer et al. [121]	2018	Gauteng	2013 and 2015	Cervical	^∨^ GP5+/6+ PCR enzyme immunoassay	Women with istologically confirmed stages of cervical disease
Mbulawa et al. [122]	2018	Gauteng	2013 and 2014	Cervical	RLA	Communities and community outreach programs
Liebenberg et al. [123]	2019	KwaZulu-Natal	N/A	Cervical	RLA	Rural and urban areas
Onywera et al. [124]	2019	Western Cape	N/A	Cervical	RLA	Gugulethu
Papasavvas et al. [125]	2019	Gauteng	N/A	Cervical	RLA	^™^ ART-treated HIV + HPV + coinfected women
Chambuso et al. [126]	2020	Western Cape	2016 and 2017	Cervical	RLA	Clinic and Hospitals
Johnson et al. [127]	2020	Western Cape	N/A	Cervical	GeneXpert HPV	Clinic
Kuhn et al. [128]	2020	Western Cape	2016 and 2016	Cervical	RLA	Community outreach and clinic
Taku et al. [129]	2020	Eastern Cape	2017 to 2019	Cervical	HCII	Clinics
Taku et al. [130]	2020	Eastern Cape	2017 to 2018	Cervical	HCII	Community health clinic
van de Wijgert et al. [131]	2020	Gauteng	2011 and 2012	Cervical	INNO-LiPA	Centres and communities
Wood et al. [132]	2020	Gauteng	N/A	Oral	Abbott RealTime	Centre and clinic
Kelly et al. [133]	2021	Gauteng	2011 to 2012	Cervical	INNO-LiPA	ART initiation site
Mbulawa et al. [22]	2021	Eastern Cape	2019	Vaginal	RLA	HPV education intervention
Onywera et al. [134]	2021	Western Cape	N/A	Cervical	RLA	HPV Couples Cohort
Saidu et al. [135]	2021	Western Cape	N/A	Cervical	GeneXpert HPV	Screening population and screening population
Taku et al. [136]	2021	Eastern Cape	2017 to 2018	Cervical	HCII	Community health clinic and the referral clinic
Taku et al. [137]	2021	Eastern Cape	2017 to 2019	Cervical	HPV direct-flow chip	Clinic
Tiiti et al. [138]	2021	Gauteng	2016 to 2018	Cervical	Abbott RealTime	Hospital
Tiiti et al. [139]	2021	Gauteng	2016 and 2018	Cervical	Abbott Realtime	Hospital
Mbulawa et al. [19]	2022	Eastern Cape	2018 to 2020	Cervical	RLA	Hospital
Naidoo et al. [140]	2022	KwaZulu-Natal	2019 to 2020	Cervical	GeneXpert HPV	Clinics
Onywera et al. [141]	2022	Eastern Cape	2019	Cervical	RLA	Schools
Tiiti et al. [142]	2022	Gauteng	2016 to 2018	Cervical	Abbott Realtime	Clinics
Botha et al. [143]	2023	N/A	N/A	Cervical	Onclarity assay	Metropolitan areas
Denny et al. [144]	2023	Western Cape	2017 to 2018	Cervical	GeneXpert HPV	Clinic
Happel et al. [145]	2023	Western Cape	2015 to 2017	Cervical	Shotgun DNA sequencing of purified virions	Youth centre
Maroga et al. [146]	2023	Gauteng	2015 to 2019	Breast tissue	RLA	Hospital
Snyman et al. [147]	2023	Gauteng	N/A	Cervical	HCII	Clinics and Hospital
Sørbye et al. [148]	2023	Gauteng and Western Cape	2016 to 2020	Cervical	Pre-Tect HPV-Proofer’7	Clinics
Vink et al. [149]	2023	Gauteng	2016 to 2020	Cervical	Cobas 4800 HPV test	Clinics
Qulu et al. [150]	2023	KwaZulu-Natal	2016 to 2017	Vaginal	RLA	Clinic
Downham et al. [151]	2024	KwaZulu-Natal	2019 to 2023	Cervical	HPV DNA Test	Centre
Habour et al. [152]	2024	Western Cape	2009 to 2019	Oral	HybriSpot HPV Direct Flow Chip	Hospital
Maueia et al. [153]	2024	Gauteng	2018 to 2019	Cervical	HybriSpot HPV Direct Flow Chip	Clinic
Sebitloane et al. [154]	2024	KwaZulu-Natal	2019	Vaginal	GeneXpert HPV	Clinic

^#^ N/A: Not indicated, ^ SBH: Southern Blot Hybridization, ^$^ ISH: In situ hybridization, ^ð^ NISH: Non-Isotopic in Situ Hybridization, * PCR: Polymerase Chain Reaction, ^α^ HCII: Hybrid Capture II test, ^&^ RFLP: Restriction fragment length polymorphism, ^€^ RT-PCR: Realtime-PCR, ^∞^ RLA: Roche Linear Array, ^°^ RLB: Roche Line Blot, ^¥^ HPV-4cLIA: HPV-4competitive Luminex Immuno Assay, ^Δ^ EIA: Enzyme immunoassay, ^©^ IBH: Line blot hybridization, ^ʓ^ SPF10 PCR-DEIA-LiPA25: Specific Primer F10 Polymerase Chain Reaction-DNA Enzyme Immuno Assay-Line Probe Assay25, ^A^ INNO-LiPA: Innovative Line Probe Assay, ^≠^ cART: combination Antiretroviral Therapy, ^∨^ GP5+/6+ PCR EIA: General Primer 5+/6+ Polymerase Chain Reaction Enzyme Immuno Assay, ^™^ ART: Antiretroviral Therapy.

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
