# Peer review of "Mapping the HPV Landscape in South African Women: A Systematic Review and Meta-Analysis of Viral Genotypes, Microbiota, and Immune Signals"

_viruses, 2024, doi:10.3390/v16121893_

Round 1
Reviewer 1 Report
Comments and Suggestions for Authors
The manuscript by Maswanganye et al. reports a systematic review and meta-analysis that evaluated HPV prevalence, genotype distribution, and associations with cervicovaginal microbiota and cytokine profiles among South African women. Results and analyses were performed from a total of 120 eligible studies, and showed higher HPV prevalence rates in cercal samples, followed by vaginal, anal and oropharyngeal samples. HPV genotypes showed results as reported elsewhere, except for the high prevalence of HPV35 in cervical samples (2nd most prevalent type). Women living with HIV had higher HPV prevalence when compared to healthy counterparts. Specific and shared bacterial taxa were associated with HIV, HPV and BV statuses, but mostly followed previous conclusions, showing a predominance of Lactobacillus in HPV-negative samples and more anaerobic bacteria in HPV+ and BV+ samples. Finally, cytokine profiles showed similar patterns, with specific and shared upregulation among HPV and HIV carriers, and downregulation of certain cytokines among HPV-negative women.
This study was very well conducted and comprehensive, as far as systematic reviews and meta-analyses are defined. Limitations (including sampling and publication biases) are well provided, and important points of concern are drawn from the data, as the need for incorporating HPV35 immunogens in an HPV vaccine for regional needs, just to cite one example. A few minor concerns, mostly of textual nature, need clarification / amendments from the authors to ensure publication of this manuscript.
1. In Figure 3, was there an effort to represent the dots inside each South African province as the specific locales of collection, or they are only representative of the province? My question is because of less dense provinces where dots are not evenly distributed (e.g., Eastern Cape). If the dots represent more specific locations, this should be stated in the legend to the Figure.
2. Figures 7 and 8: Although I understood that the authors wished to display the HPV genotypes in a numerical order, this is not the way pie charts are usually represented. They rather should show in the order from the most to the least prevalent type.
3. The new paragraphs of the manuscript (those added to the second version recently distributed to reviewers) related to Study Quality and Bias Assessment (Methods) and Risk of Bias Assessment (Results) should be revised by a native English speaker, as they contain several grammar and syntax errors.
4. Cosmetic remark: please change “shirt” in line 571 to “shift”.
Comments on the Quality of English LanguagePlease see item no. 3 above.
Author Response
General response to Reviewer 1:
We sincerely appreciate your thorough review and insightful comments, which have greatly contributed to improving the quality of our manuscript. We have carefully addressed all your suggestions and concerns and made the necessary revisions to enhance clarity and rigor. Detailed responses to each of your comments are provided below. Thank you for your valuable feedback and guidance.
Comment 1: In Figure 3, was there an effort to represent the dots inside each South African province as the specific locales of collection, or they are only representative of the province? My question is because of less dense provinces where dots are not evenly distributed (e.g., Eastern Cape). If the dots represent more specific locations, this should be stated in the legend to the Figure.
Response:
The comment is well acknowledged. Yes, each dot inside each South African province represents each study, were a different sample type represented in the different dot colour was used. Figure 3 was edited with equal dot sizes in each province to avoid ambiguity. Additionally, the figure legend was modified for clarity. See below legend;
“Figure 3: Geographic Distribution of Sample Types Used in HPV Studies Across South African Provinces. The map illustrates the geographic distribution of various sample types used in HPV studies across South Africa. Each dot represents an individual study, and the colour of the dots indicates the sample type: pink for cervical, green for vaginal, yellow for oral, red for anal, blue for breast tissue, brown for cervical and oral, black for laryngeal, turquoise for cervical and blood, and purple for biopsy. The position of each dot within a province does not necessarily reflect the exact location of the study within that province. The map highlights key regions, such as Gauteng and the Western Cape, where HPV research has been most prominent. The Gauteng province, located in the top-right corner, is zoomed out to ensure all sample dots are visible.”
This edited section is in page 8, lines 397-408.
Comment 2: Figures 7 and 8: Although I understood that the authors wished to display the HPV genotypes in a numerical order, this is not the way pie charts are usually represented. They rather should show in the order from the most to the least prevalent type.
Response:
The comment was well received. As the reviewer recommended, we updated figure 7 and 8 to show the order from the most to the least prevalent HPV type. This is in pages 13-1, lines 537-408
Comment 3: The new paragraphs of the manuscript (those added to the second version recently distributed to reviewers) related to Study Quality and Bias Assessment (Methods) and Risk of Bias Assessment (Results) should be revised by a native English speaker, as they contain several grammar and syntax errors.
Response:
The comment is acknowledged. The sections on Study Quality and Bias Assessment of the methods and Risk of Bias Assessment of the Results were revised to improve the language and grammar. This is in page 4-5, lines 185-289.
Comment 4: Cosmetic remark: please change “shirt” in line 571 to “shift”.
Thank you for the comment. The word shirt in line 571 was changed to shift now in page 16, line 721.
Final remarks
Additionally, the manuscript was proofread by all the authors for enhancing the quality and Grammar.
Reviewer 2 Report
Comments and Suggestions for Authors
All abbreviation should be removed from Abstract. The study is of great interest who is working with HPV. The interaction between HIV and HPV has been well demonstrated. A lot of different results have been achieved. Therefore, many questions arise regarding the content of the manuscript, which the authors could resolve in future studies. Based on the data obtained, is it possible to get answers why there are so many types of HPV and what affects their number: stability to environmental conditions, differences in the structure of viruses or their DNA/proteins, geographical reasons, etc. Could these simply be transitional forms in the process of aging or maturation of viruses? There are not enough conclusions or conclusions for each paragraph so that a logical connection between paragraphs or subtopics is visible. Therefore, the extent of interaction with microbiota is completely unclear. As is known, the microbiota in uninfected mammals forms from 70 to 100% of immunity. Gnotobiotic organisms, that is, absolutely sterile, have no immunity at all. This is not mentioned in the work. It would be more emphasized to use interferon as an indicator of the generation of immunity, which in the chain of events: activation of the immune response -----> interferon ---> chemokines ---> comes before interleukins. Further, about 360 types of papillomaviruses are known, approximately 200 types in mammals. How can the authors predict vaccine design for 200 types of papillomaviruses? Most likely, it is necessary to analyze the epitopes, select the most common ones, and only on their basis can a presumptive variant of the genetic design for the vaccine be made. Also not discussed is one of the most mysterious aspects of HPV - the presence of hidden forms, lacunae, etc., which can help avoid an immune attack and survive without damage for subsequent generations and the replication of intact offspring of the virus.
Author Response
General response to Reviewer 2:
We appreciate the reviewer’s insightful comments; however, we would like to clarify that the main objective of our study was to systematically review HPV research in South Africa and conduct a meta-analysis to better understand the prevalence and regional distribution of HPV genotypes among South African women. Additionally, we examined how HIV status influences HPV infections, as well as the roles of cytokine profiles and cervicovaginal microbiota in HPV prevalence and progression. Given the focus of our study, we did not intend to explore the broader evolutionary aspects of HPV (such as the reasons for the diversity of HPV types or the latent forms of HPV), as these topics are outside the scope of this review. Detailed responses to each of the reviewer’s points/comments are provided below. Thank you for your valuable feedback and guidance.
Comment: All abbreviation should be removed from Abstract.
Response:
We acknowledge the reviewer’s comment. Therefore, all abbreviations were written in full at first mention then abbreviated throughout the document.
Comment: “Is it possible to get answers why there are so many types of HPV and what affects their number: stability to environmental conditions, differences in the structure of viruses or their DNA/proteins, geographical reasons, etc. Could these simply be transitional forms in the process of aging or maturation of viruses? “
Response:
We appreciate the reviewer’s point on the diversity of HPV types and influencing factors. However, our study focused specifically on HPV prevalence and genotype distribution in South Africa, particularly in relation to HIV co-infection, cytokines, and microbiota. Broader discussions on HPV evolution were beyond the scope of our analysis, and will be incorporated in our future studies.
Comment: “Therefore, the extent of interaction with microbiota is completely unclear. As is known, the microbiota in uninfected mammals forms from 70 to 100% of immunity. Gnotobiotic organisms, that is, absolutely sterile, have no immunity at all. This is not mentioned in the work.”
Response:
We recognize the importance of microbiota in immune responses. However, the focus of our study was on cervicovaginal microbiota and its association with HPV prevalence and progression in the South African cohort. Our review specifically examined how cervicovaginal microbiota interacts with HPV in this population, which is a narrower aspect of broader microbiota research. We did not explore microbiota’s general role in immunity, as this was beyond the scope of our meta-analysis.
Comment: “It would be more emphasized to use interferon as an indicator of the generation of immunity, which in the chain of events: activation of the immune response -----> interferon ---> chemokines ---> comes before interleukins. “
Response:
While we recognize the importance of interferons in the immune response, our study did not extensively explore cytokine pathways, such as interferon activation or the cascade involving chemokines and interleukins. Instead, our review focused on the role of cytokines and immune factors in HPV prevalence and progression, particularly in the context of HIV co-infection.
The inclusion of specific immune pathways like interferons may be relevant for future research, but it was outside the scope of our review. We also mention in our manuscript that a limitation of current cytokine studies is the variability in the cytokines tested, which may hinder our understanding of the HPV-immune system associations. Additionally, indicated as another limitation of the study is that there is limited number of researches in South Africa that investigated the interplay between HPV and cytokine profiles.
Comment: “Further, about 360 types of papillomaviruses are known, approximately 200 types in mammals. How can the authors predict vaccine design for 200 types of papillomaviruses? Most likely, it is necessary to analyze the epitopes, select the most common ones, and only on their basis can a presumptive variant of the genetic design for the vaccine be made.”
Response:
The reviewer’s comment on vaccine design for the 200+ HPV types is relevant to broader HPV research but falls outside the focus of our study. Our review aimed to contribute to the development of vaccines targeting cancer-causing HPV types, with a specific focus on the prevalence and regional distribution of HPV genotypes in South Africa and the impact of HIV on HPV infections. We did not aim to predict the development of a universal vaccine, as this requires a different research approach centred on epitopes and immunology.
Comment: “Also not discussed is one of the most mysterious aspects of HPV - the presence of hidden forms, lacunae, etc., which can help avoid an immune attack and survive without damage for subsequent generations and the replication of intact offspring of the virus.”
Response:
The issue of latent or hidden HPV forms is an interesting area of ongoing research. However, it falls outside the scope of our systematic review and meta-analysis, which did not focus on this topic.
Final Remarks:
We appreciate the reviewer’s comments, which have provided us with valuable insights and ideas for potential future research